# A Comparative Study of Inhibition Function between High-Intensity Interval Training and Moderate-Intensity Continuous Training in Healthy People: A Systematic Review with Meta-Analysis

**DOI:** 10.3390/ijerph20042859

**Published:** 2023-02-06

**Authors:** Qianqian Wu, Xiaodan Niu, Yan Zhang, Jing Song, Aiping Chi

**Affiliations:** School of Sports, Shaanxi Normal University, Xi’an 710119, China

**Keywords:** high-intensity interval training, moderate-intensity continuous training, executive function, inhibition function, healthy people

## Abstract

Meta-analysis was used to compare the effects of two interventions, high-intensity interval training (HIIT) and moderate-intensity continuous training (MICT), on inhibition in executive function in healthy people, providing some theoretical basis for exercise practice and health interventions. We searched the PubMed, Science Direct, Web Of science, Cochrane, and CNKI databases for relevant articles on the inhibition function effects of HIIT and MICT in healthy populations for the period of library construction to 15 September 2022. The basic information of the screened literature was organized and summarized using Excel. Statistical analysis of the correct rate and response time indicators of the inhibition function in the HIIT and MICT groups was performed using Review Manager 5.3 analysis software. A total of 285 subjects from 8 studies were included in this study, the number of HIIT subjects was 142, the number of MICT subjects was 143, including teenagers, young adults, and the elderly. Eight studies included response time, and four included correct rate and response time. The standardized mean difference (SMD) for correct rate inhibition function in the HIIT and MICT groups was 0.14, 95% CI (−0.18, 0.47), SMD at response time was 0.03, 95% CI (−0.20, 0.27). In addition, no significant differences were found between the two exercise modalities in either the intervention period or the population receiving the intervention. Both HIIT and MICT could improve inhibition function in healthy people, but there was no significant difference in the improvement effect between them. It is hoped that this study can provide some references for people’s choice of health intervention methods and clinical practice.

## 1. Introduction

Executive function refers to the process of self-control and coordination by using the higher cognitive functions in the human brain when people encounter difficulties or want to achieve certain goals, mainly including inhibition, refreshment, and transformation functions [1]. Good executive function is important for ensuring healthy development. Conversely, disruptions in executive function can lead to cognitive deficits, emotional disturbances, and other adverse physiological responses [2]. Inhibition function, also known as inhibition control, helps people to focus on a task by excluding information that is not relevant or distracting. This allows them to concentrate more on the task at hand and resist urges or impulses [3]. If the inhibition efficiency decreases, it will lead to a decrease in the efficiency of task completion. This is because the ability to resist external temptations will be reduced, and more interfering information will enter the working memory [4]. The development of inhibition is a topic of cognitive psychology which has demonstrated to have crucial role in abilities such as learning [5], social–emotional aspects [6], and self-regulation [7]. Adolescence is a pivotal time for brain development, and making a concerted effort to improve inhibition function during this stage will have far-reaching consequences [8]. The degeneration of tissues associated with aging often leads to a decline in cognitive function as we age. However, inhibition function plays an important role in life [9], so improving inhibition in adults could also reduce the burden on society.

Inhibition function plays an important role in the executive functions of the brain [10], one study included 4304 adolescents aged 6 to 12 years and then analyzed the effect of sedentary time on brain executive function in healthy adolescents, showing that children with long sedentary time and low physical activity levels had the highest brain executive function scores and both brain function produced disorders [11]. The new coronavirus epidemic has resulted in a significant increase in sedentary time among people, with the greatest increase occurring in adolescents [12]. Several studies have indicated that exercise interventions can change the way major brain regions such as the motor cortex and cerebellum are activated, and improve the connections between these regions. This can have a positive effect on people’s inhibitory abilities [13]. Jiang’s study found that moderate soccer training for 8 weeks resulted in significant improvements in executive function in children, specifically in inhibition function [14]. A meta-analysis showed that aerobic exercise and physical and mental exercise had positive effects on working memory, cognitive flexibility, and inhibition function in healthy older adults [15]. However, some experiments have also shown that motor interventions do not have a significant improvement in inhibition function in adults [16]. This may be due to the rapid brain development and the high plasticity of neural networks in children, whereas in adults the prefrontal cortical neuronal cells are more developed, and also related to the duration of the intervention and the intensity of the intervention.

HIIT is an effective training method that involves repeatedly performing short periods of high-intensity exercise, with active recovery at each interval, this produces more intense stimulation of the cardiovascular and muscular systems [17]. MICT exercises are of lower intensity and last longer [18]. HIIT has gained popularity in recent years due to its shorter duration and higher energy expenditure, as compared to MICT [19]. Exercise has been shown to improve inhibition by increasing cerebral blood flow [20], promoting the secretion of brain-derived neurotrophic factors [21], and improving activation patterns in brain regions such as the prefrontal and cingulate gyrus [22]. Kao observed that [23], after a single session of HIIT and MICT exercise, subjects showed improved performance on the flanker task following HIIT exercise, as compared to MICT. The findings of a study by Costsee indicated that [24] the MICT group was significantly more effective than the HIIT group in terms of improving executive function in older adults. Zhang discovered that [25] adolescents’ response times decreased after both HIIT and MICT interventions in comparison to pre-exercise, but there was no significant disparity between the two groups.

Although some research has been conducted comparing the effects of HIIT and MICT methods on the improvement of inhibition function in executive functions in healthy individuals [23,24,25], a more systematic and quantitative evaluation is needed. Based on this, this study used a systematic review with meta-analysis to quantitatively summarize the latest literature and compare the effects of HIIT and MICT on the improvement of human brain inhibition function to provide a theoretical basis for future exercise practice and health interventions.

## 2. Methods

### 2.1. Search Strategy

In this study, two researchers searched PubMed, Science Direct, Web of Science, Cochrane, and CNKI databases by computer, using a hybrid search combining subject terms and free words in English and Chinese. The search terms were mainly (High-Intensity Interval Training or High Intensity Interval Training or High-Intensity Interval Trainings or Interval Training, High-Intensity or Interval Trainings or High-Intensity or Training, High-Intensity Interval or Trainings, High-Intensity Interval or High-Intensity Intermittent Exercise or Exercise, High-Intensity Intermittent or Exercises, High-Intensity Intermittent or High-Intensity Intermittent Exercises or Sprint Interval Training or Sprint Interval Trainings) and (moderate-intensity continuous exercise) and (Executive Function or Executive Functions or Function, Executive or Functions, Executive or Executive Control or Executive Controls or Inhibitory control). To make the search more comprehensive, we also conducted a manual search from published meta-analyses to literature reviews. The search was conducted from the date of creation to September 15, 2022.

### 2.2. Selection Criteria

Study screening was conducted by two pretrained researchers, who screened the studies retrieved from each database separately according to inclusion and exclusion criteria, and then pooled the selected studies. If the conclusions of the two researchers did not agree, the disagreement was first resolved through discussion. If the problem remained unresolved, a third-party expert adjudicated and agreement was obtained before inclusion.

The inclusion criteria were as follows: (1) All studies published in the public domain; (2) the interventions used in the experiment must include HIIT and MICT; (3) the control measures were randomized controls; (4) the experimental subjects were healthy people; (5) the tests used for inhibition control were go/no go, flank, and Stroop methods, and the outcome indicators were response time and correct rate.

The exclusion criteria were as follows: (1) Exclusion of review articles, conference abstracts, animal experiments, and other articles for which full text and outcome indicators are not available; (2) experimental interventions other than HIIT and MICT, or only one of these interventions; (3) the experimental design did not meet the requirements of this study; (4) the measurement paradigm used or the outcome indicators did not meet the requirements of this study; (5) subjects with chronic diseases, mental illnesses, etc.

### 2.3. Data Extraction

According to the needs of the study, two researchers were arranged to independently extract and enter data after reading the full text, which mainly included the following: First author of the article; year of publication; age and number of participants; intervention method (intervention content, intensity, duration, frequency, etc.); test methods and results of outcome indicators. Ending indicators were entered according to the Cochrane handbook of systematic evaluation, and the difference between the mean and standard deviation before and after the intervention was extracted and, if not reported in the literature, calculated by the following formula: Mean (change) = Mean (post) − Mean (pre), SD (change) = SQRT [(SD^2^ (post) + SD^2^ (pre)) − (2 × corr × SD (post) × SD (pre))], corr set to 0.5. If only the standard error (SE) and 95% confidence interval are provided in the literature, the SD = SE × SQRT (N), where N is the sample size, or SD = SQRT (N) × [(UCI − LCI)/3.92], where U = upper CI and L = lower CI [26].

### 2.4. Quality Assessment

The methodological quality of the included studies was assessed by two researchers using the Cochrane risk of bias assessment tool, which includes six items: Compliance with random assignment; whether the assignment scheme was concealed; whether blinding was used; whether the outcome data were complete and whether there were cases of shedding; whether the study results were selectively reported; and whether there were other sources of bias. According to the Cochrane handbook, each evaluation is divided into three levels of “low risk”, “unclear”, and “high risk” [27], and researchers are required to evaluate the levels according to the actual situation, and if there is disagreement among researchers, it is determined through consultation with third-party experts. Satisfying ≥5 items is considered low risk of bias, 3–4 items are considered moderate risk of bias, and <3 items are considered high risk of bias [28].

In addition, this research also applied the Physiotherapy Evidence Database (PEDro) scale to evaluate the methodological quality of the included studies [29,30].

### 2.5. Statistical Methods

The basic information and data of the acquired study were organized and summarized by Excel 2013, and meta-analysis of the two types of outcome indicators included in this study was performed by applying Review Manager 5.3 software. Due to the differences between the measurement tasks selected for this study and the experimental subjects, standardized mean difference (SMD) was used to express effect sizes and to estimate 95% confidence intervals, and *p* < 0.05 indicated that the study was statistically significant. Heterogeneity among the included literature was tested using a Q test and I^2^ test, *p* > 0.1 and I^2^ < 50% indicate homogeneity of the included indicators in this study, and meta-analysis should be performed using a fixed effect model, otherwise meta-analysis should be performed using a random effect model [19]. The size of heterogeneity was measured by I^2^, with I^2^ < 25%, 25–49%, 50–74%, ≥ 75% representing low, moderate, high, and very high heterogeneity, respectively. The amount of effect is classified as: 0.2–0.5, 0.5–0.8, >0.8, representing small, medium, and large effect, respectively. When heterogeneity is significant, subgroup analysis or sensitivity analysis is required. A funnel plot and Egger method were used to check whether publication bias existed.

## 3. Results

### 3.1. Results of the Literature Search

The results of the study search and screening from the database are shown in Figure 1. A total of 856 studies were retrieved, and 814 studies remained after removing duplicates. Seven hundred and ninety-nine studies that were not relevant to this study were removed by reading the titles and abstracts, and finally fifteen studies were screened by reading the full text, among which three studies were not RCT studies and four studies did not provide the original data of the outcome indicators, and finally we performed meta-analysis on eight of them.

### 3.2. Risk of Bias Estimation

The quality of the eight RCTs included was assessed methodically using two methods, all of which provided detailed descriptions of subjects’ inclusion conditions and described the use of random assignment to group them, but only one study provided detailed description of assignment [25]. Subject blindness was mentioned in one study [25], while subject, healer, and evaluator blindness was not mentioned in other studies. All studies provided complete measurement data and performed statistical analysis. The results of studies’ quality evaluation are shown in Figure 2 and Table 1.

### 3.3. Research Characteristics

Basic information on the inclusion of all studies is shown in Table 2. Two hundred and eighty-five study subjects were involved in eight randomized controlled studies, all of which included an outcome indicator response time. The interventions were HIIT and MICT, with publication years from 2017–2021. Three studies [31,35,36] were acute interventions. Five studies were long-term interventions [24,25,32,33,34] with a frequency of three times/week, two studies had a six-week intervention period [32,33], two studies had an eight-week intervention period [25,34], and one study had a sixteen-week intervention period [24]. Five studies [24,25,33,35,36] had a running mode of exercise, two studies had cycling [31,32], and one study had a combination exercise [34]. Two studies were carried out in adolescents [25,34], five studies in middle-aged and young adults [31,32,33,35,36], and one study in the elderly [24]. There is one study using the Flanker task [34] and seven studies using the Stroop task [24,25,31,32,33,35,36]. Four studies [24,25,34,36] included correct outcome indicators, one study [36] for acute interventions and three studies [24,25,34] for long-term interventions.

### 3.4. Results of Meta-Analysis

#### 3.4.1. Correct Rate Results

Meta-analysis of correct rate after HIIT and MICT interventions was performed, and the test showed that *df* = 3, *p* = 0.24, *I*^2^ = 29% < 50%, suggesting a low heterogeneity in the literature for this inclusion, and a fixed effects model was chosen for the meta-analysis. Combined effect size: *SMD* = 0.14, *95% CI* (−0.18, 0.47), *Z* = 0.86, *p* = 0.39. The forest plot (Figure 3) showed that the horizontal line of the 95% CI for the indicator SMD in the HIIT and MICT groups was in the middle of the null line, indicating that there was no significant difference between the results of the two groups.

#### 3.4.2. Response Time Results

Meta-analysis of response time after HIIT and MICT interventions was performed, and the test showed that *df* = 7, *p* = 0.50, *I*^2^ = 0% < 50%, suggesting that there was no heterogeneity in the literature included in this study, so a fixed effects model was chosen for the meta-analysis. Combined effect size: *SMD* = 0.03, *95% CI* (−0.20, 0.27), *Z* = 0.28, *p* = 0.78. The forest plot (Figure 4) showed that the horizontal line of the 95% CI for the indicator SMD in the HIIT and MICT groups was in the middle of the null line, indicating that there was no significant difference between the results of the two groups.

#### 3.4.3. Results of Subgroup Analysis

The effect of HIIT and MICT on inhibitory function may be influenced by the duration of the intervention and age size, so this study was conducted with subgroup analysis for different intervention periods and different intervention populations.

According to the results of the subgroup analysis of the intervention method, as only one study [36] included correct rate after acute exercise, we performed a meta-analysis of correct rate after long-term exercise, which showed no significant difference in the improvement of correct rate after long-term HIIT and MICT interventions in the healthy population (*SMD* = 0.32, *95% CI* (−0.06, 0.70), *p* = 0.53, *I*^2^ = 0%, Figure 5). There was no significant difference between acute HIIT and MICT interventions for improvement in response time (*SMD* = −0.05, *95% CI* (−0.56, 0.46), *p* = 0.39, *I*^2^ = 55%, Figure 6). There was no significant difference in the improvement in response time after long-term intervention (*SMD* = 0.03, *95% CI* (−0.20, 0.27), *p* = 0.78, *I*^2^ = 0%, Figure 6).

The results of the subgroup analysis of the intervention population showed no significant difference between HIIT and MICT for the improvement of correct rate in adolescents (*SMD* = 0.27, *95% CI* (−0.20, 0.74), *p* = 0.29, *I*^2^ = 11%, Figure 7). There was no significant difference in the improvement of the correct rate for adults (*SMD* = 0.03, *95% CI* (−0.72, 0.78), *p* = 0.13, *I*^2^ = 57%, Figure 7). There was no significant difference in the improvement in response time for adolescents (*SMD* = −0.03, *95% CI* (−0.46,0.41), *p* = 0.49, *I*^2^ = 0%, Figure 8). There was no significant difference in the improvement in response time for adults (*SMD* = 0.06, *95% CI* (−0.22, 0.33), *p* = 0.33, *I*^2^ = 13%, Figure 8).

### 3.5. Publication Bias

In this study, the funnel plot and Egger method were used to test for the presence of publication bias. The funnel plots (Figure 9 and Figure 10) showed that the scatter of each study effect showed a relatively symmetrical inverted funnel shape, indicating that the possibility of publication bias in the study included in this search was low. Egger’s method test showed: correctness: *t* = −0.28, *p* = 0.803 > 0.05, *95% CI* (−19.72, 17.28) and *t* = 0.25, *p* = 0.814 > 0.05, *95% CI* (−4.63, 5.67) at response, further indicating that there was no publication bias in the literature included in this study.

## 4. Discussion

The aim of this study was to directly compare the effects of HIIT and MICT on the improvement of inhibition function in a healthy population through quantitative analysis using the meta-analysis method. Our results showed no significant differences in the improvement of inhibition function between the two exercise modalities overall, and no significant differences in the target population and duration of the intervention.

Compared to MICT, HIIT has a variety of types, and shortens the duration of exercise, but the exercise effect does not diminish, and even shows better physiological adaptations, which is ideal for people with busy work schedules and lack of exercise time [37], and is becoming increasingly popular among the general public. Many scholars have included HIIT in research on healthy people. HIIT has been shown to be effective in improving inhibition function in children in multiple studies, and can be sustained for longer periods of time [38]. Alves performed a study of short one-time HIIT exercises for executive function in healthy middle-aged men and showed a significant reduction in reaction time to perform the Stroop task after exercise [39]. This may be due to the fact that HIIT reduces glucose uptake and increases brain lactate metabolism, and the increased brain lactate content can be used as a source of energy for cognitive activities [40,41]. HIIT can also more effectively promote the body’s subsecretion of BDNF [42], irisin [43], and insulin-like growth factor I [44], which in turn can contribute to neurogenesis, enhance the interaction between brain neurological functions, and improve brain neuroplasticity and cognitive function. Furthermore, the vomeronasal nucleus located in the prefrontal cortical striatal system aids in inhibition function processes, and exercise initiation can stimulate dopamine receptors in the vomeronasal nucleus region, thereby moderating inhibitory response control [45]. The elevation of hormones such as norepinephrine and neuropeptide Y caused by exercise also has an effect on inhibition function [46].

The many advantages of HIIT have caused it to receive more scrutiny, but there is no clear consensus on which has a more significant effect on inhibition function between HIIT and MICT. Yang showed the effects of single HIIT and MICT exercise interventions on executive function in 36 university students and showed that MICT had a significantly better effect on inhibition function than the HIIT group [47]. Cai administered an exercise intervention to children for 8 weeks, and although the HIIT group had a higher response time to inhibition function than the MICT group, there was no significant difference [48]. Tian found that acute HIIT was more effective than MICT in increasing inhibition function in young people, lasting for about 90 min [49]. Lambrick found that blood flow and oxygenation in the prefrontal cortex increased after intermittent exercise and MICE, but inhibition function improved only after intermittent exercise [50]. The role of HIIT on the improvement of inhibition function is still being explored and more experiments are needed for a more in-depth understanding.

This study also conducted subgroup analyses of the duration of intervention for HIIT and MICT, but there were no significant differences between acute and long-term exercise. Acute exercise may affect brain regions such as the anterior cingulate gyrus and prefrontal cortex associated with inhibitory function and increase cerebral blood flow and brain-derived neurotrophic factor concentrations contributing to cognitive function [51]. Long-term exercise has been shown to improve overall cognitive function [52] and enhance cardiovascular function more effectively [53]. Studies have shown that adolescents who engage in HIIT and MICT for six weeks experience significant improvements in inhibition function, with HIIT being more effective [54]. However, some studies have also shown that long-term HIIT did not significantly improve inhibition function and did not play a larger role in changing proBDNF concentrations [15]. A meta-analysis also showed that long-term physical exercise did not have a significant effect on executive function [34]. This may be due to the increased differences in time span and changes in the degree of cooperation of the study subjects that influenced the effect of the intervention. Although our study also showed no significant differences between exercise cycles, long-term exercise brings many benefits to the body, and in the future we will pay more attention to the effect of exercise duration on inhibition function.

In addition, this study noted the large age range of subjects in the included studies, so a subgroup analysis of the intervention population was performed. The results showed that there was no significant difference in the effect of HIIT and MICT intervention on adolescents and adults. Zhang [25] found that the response time and accuracy of adolescents in the two groups after HIIT and MICT intervention decreased compared with that before exercise, and the difference between groups was not significant. Zhang [54] showed that after HIIT and MICT intervention, the physical fitness and inhibition ability of children around 10 years old were improved compared with that before exercise, and the effect of HIIT was more significant. Compared with the traditional continuous exercise, HIIT shows great changes in the type of exercise, intensity, speed, and other aspects, which are more in line with the physical and mental development characteristics of teenagers, avoiding the monotonous and boring training mode, and its intervention effect may be better. For adults, the effects of the two types of exercise intervention are not uniform. Fang [55] showed that HIIT had a more significant effect on improving cognitive function in the elderly, but Brown [31] and Costsee [24] showed that the intervention effect of MICT was better than HIIT. Although HIIT has many advantages, not all age groups can accept its high intensity on the body, especially the elderly population, and there is a certain risk in the implementation of HIIT intervention.

HIIT, as an emerging method of sports training, has been favored by people in recent years. Compared with the traditional continuous exercise, HIIT has distinct features such as interest and timeliness, which can improve people’s sense of pleasure in the process of exercise [56]. Studies have shown that HIIT can regulate the content of animal brain-derived neurotrophic factor (BDNF), improve the function of neurons, and reduce and increase central nervous plasticity [57]. Although HIIT has significant advantages, due to its long-term use in the field of competitive sports, HIIT has been slowly carried out in the general population in recent years, and its safety has yet to be verified, especially for middle-aged and elderly people. As their cardiopulmonary function is gradually declining, compared with the high intensity of HIIT, continuous aerobic exercise is less risky. At present, there are few studies directly comparing HIIT and MICT on the improvement of cognitive function, and the results are controversial. Therefore, a quantitative analysis method is used in this study for comparison, and the results show that there is no significant difference between the two forms of exercise, which provides some references for people to choose the forms of exercise in the future.

There are some shortcomings in this study. First, this study was not registered on PROSPERO beforehand, and we will pay more attention to this requirement in future studies and actively seek experts in related fields to control the details of the article. Second, there are multiple experimental paradigms to test the suppression function, and since we do not fully unify the paradigm requirements, whether using different paradigms will have an impact on the results needs to be further verified. Third, although we have tried our best to find the literature related to this study, there is still a possibility of omission and the quality of the included literature is mostly moderate, which may have some influence on the reliability of the meta-analysis. Fourth, most studies did not provide information about the subjects’ daily physical activity level, so we did not make a summary of this in our study. The subjects’ daily physical activity level may affect the final test result. Fifth, whether the effects of the two exercise interventions on inhibitory function are related to gender is also worthy of future detailed discussion. Finally, due to the limitation of data in the literature, the analysis and discussion of the content of the subgroup analysis are not comprehensive and in-depth, and more perspectives are needed in the future to explore the effect of exercise on the improvement of inhibition function.

## 5. Conclusions

The results of a meta-analysis showed that there is no significant difference between HIIT and MICT in terms of improving inhibition function in healthy individuals. Each type of exercise has its own advantages, so different people can choose the one that suits their needs and preferences. Future studies could further analyze the effects of both exercise modalities on interventions for other components of executive function, as well as include studies with special populations.

## Figures and Tables

**Figure 1 ijerph-20-02859-f001:**
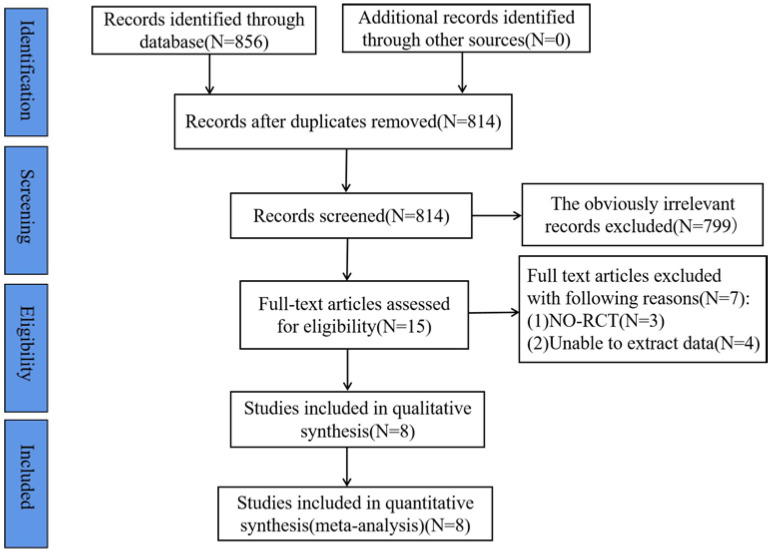
Flow chart of literature search.

**Figure 2 ijerph-20-02859-f002:**
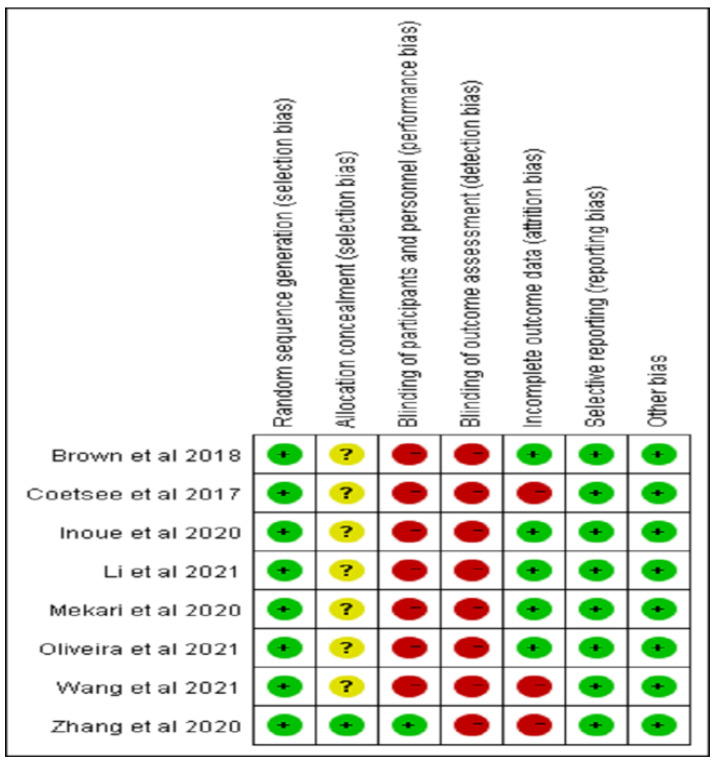
Schematic diagram of risk of bias [24,25,31,32,33,34,35,36]. Note: Green: Low risk of bias; Yellow: Unclear risk of bias; Red: High risk of bias.

**Figure 3 ijerph-20-02859-f003:**
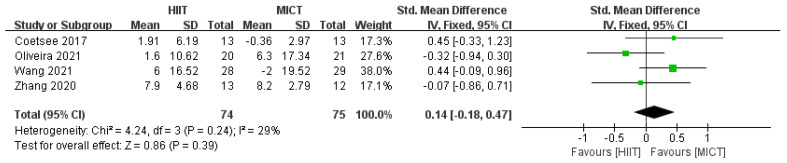
Effect of HIIT and MICT interventions on correct rate [24,25,34,36].

**Figure 4 ijerph-20-02859-f004:**
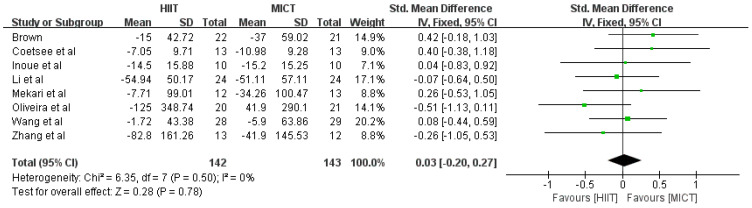
Effect of HIIT and MICT interventions on response time [24,25,31,32,33,34,35,36].

**Figure 5 ijerph-20-02859-f005:**
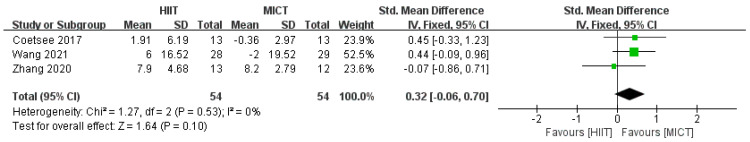
Subgroup analysis of HIIT and MICT on the correct rate of intervention period [24,25,34].

**Figure 6 ijerph-20-02859-f006:**
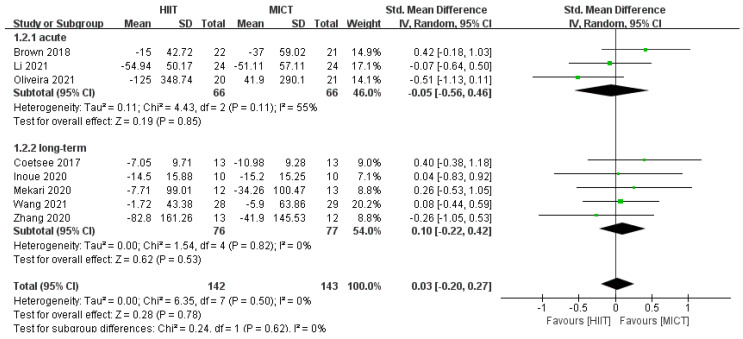
Subgroup analysis of HIIT and MICT in response time to the intervention period [24,25,31,32,33,34,35,36].

**Figure 7 ijerph-20-02859-f007:**
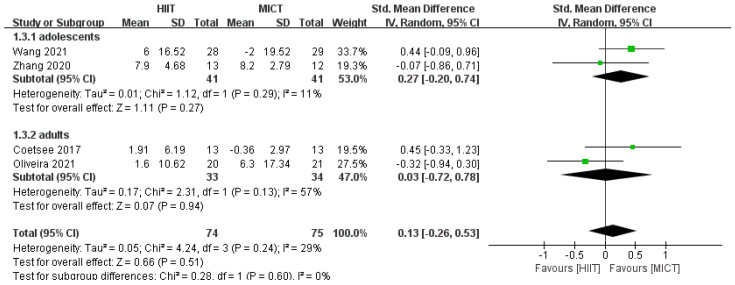
Subgroup analysis of HIIT and MICT for correct rate in the intervention population [24,25,34,36].

**Figure 8 ijerph-20-02859-f008:**
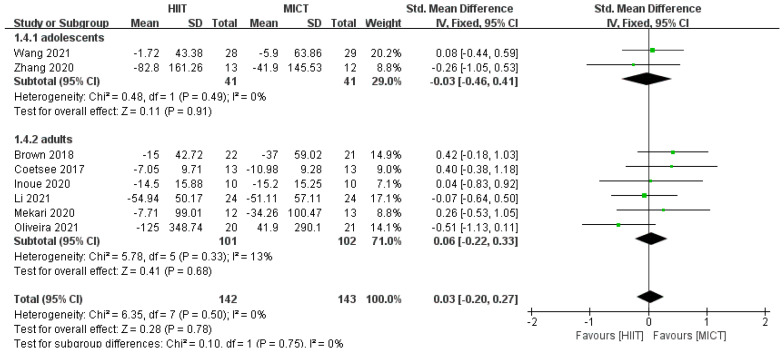
Subgroup analysis of HIIT and MICT on the response time of the intervention population [24,25,31,32,33,34,35,36].

**Figure 9 ijerph-20-02859-f009:**
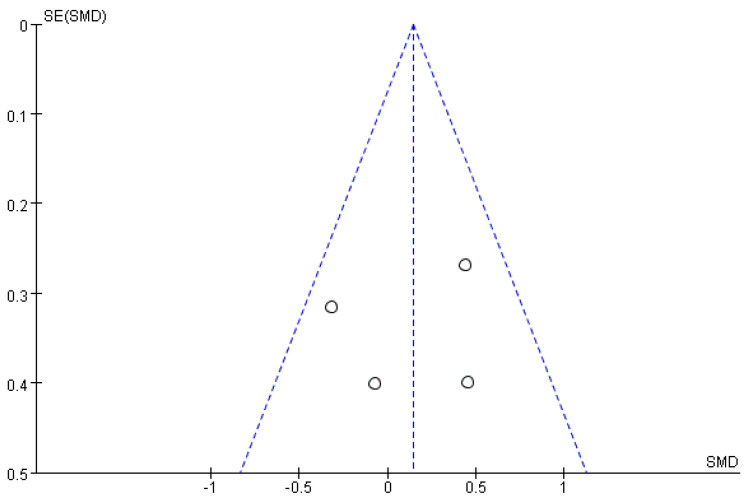
Funnel plot of publication bias for correct rate.

**Figure 10 ijerph-20-02859-f010:**
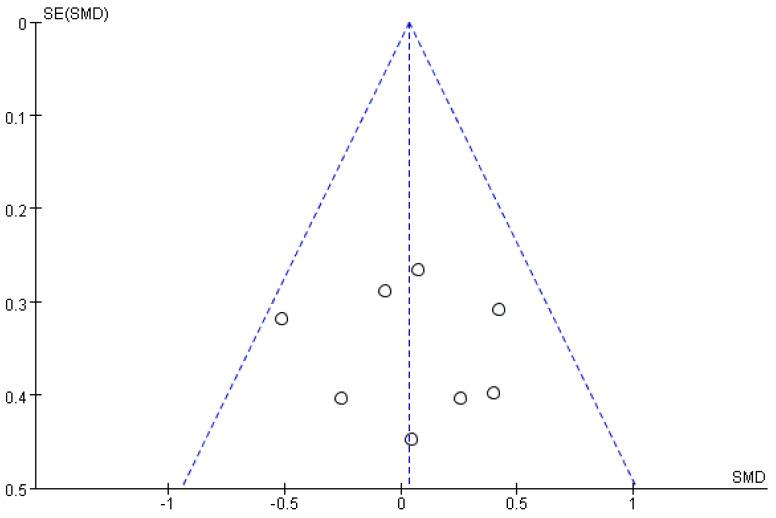
Funnel plot of publication bias for response time.

**Table 1 ijerph-20-02859-t001:** PEDro Scale scores included in the study.

Study	1	2	3	4	5	6	7	8	9	10	11	Total Score	Quality
Coetsee 2017 [24]	Yes	Yes	No	Yes	No	No	No	No	No	Yes	Yes	4	Medium
Brown 2018 [31]	Yes	Yes	No	Yes	No	No	No	Yes	No	Yes	Yes	5	Medium
Zhang 2020 [25]	Yes	Yes	Yes	Yes	Yes	No	No	No	No	Yes	Yes	6	High
Mekari 2020 [32]	Yes	Yes	No	Yes	No	No	No	Yes	No	Yes	Yes	5	Medium
Inoue 2020 [33]	Yes	Yes	No	Yes	No	No	No	Yes	No	Yes	Yes	5	Medium
Wang 2021 [34]	Yes	Yes	No	Yes	No	No	No	Yes	No	Yes	Yes	5	Medium
Li 2021 [35]	Yes	Yes	No	Yes	No	No	No	Yes	No	Yes	Yes	5	Medium
Oliveira 2021 [36]	Yes	Yes	No	Yes	No	No	No	Yes	No	Yes	Yes	5	Medium

Note: Contents of each entry: 1. Clear inclusion conditions; 2. Random allocation; 3. Distribution hiding; 4. Similar baseline indicators; 5. The subjects were blinded; 6. The therapist is blind; 7. Evaluators blind; 8. Adequate follow-up (more than 85% of subjects completed the test); 9. Intention processing and analysis; 10. Comparison between groups; 11. Point estimates and variability. Yes = 1, No = 0, no score for the first rule; 9~10, very high quality; 6~8, high quality; 4~5, medium quality; ≤3, low quality.

**Table 2 ijerph-20-02859-t002:** Basic information of the included studies.

Author	Year	Intervention Methods	Sample Size	F/M	Age	Intervention Programs	Intervention Frequency	Intervention Period	Ending Indicators
Coetsee [24]	2017	HIIT	13	3/10	64.5 ± 6.3	Warm-up; then 4 min intervals on the treadmill at an exercise intensity of 90–95% HRmax, with 3 min active recovery periods at an exercise intensity of 70% HRmax, lasting about 30 min; recovery.	3 times/week	16 weeks	Stroop b, a
		MICT	13	3/10	61.6 ± 5.8	Warm up; then run continuously on the treadmill for 47 min at an exercise intensity of 70–75% HRmax; recovery.			
Brown [31]	2018	HIIT	22	-	19.95 ± 1.96	1 min exercise intensity of 70% PPO power cycling (90–120 rpm), followed by 1 min exercise intensity of 12.5% PPO power cycling (60–90 rpm), for a total of 10 sets.	1 time		Stroop b
		MICT	21	-	20.14 ± 1.90	Continuous power cycling exercise at a fixed exercise intensity of 41.25% PPO (60–90 rpm).			
Zhang [25]	2020	HIIT	13	6/7	12.77 ± 0.44	Warm-up 5 min; 1 min running (exercise intensity about 90% HRmax), 2 min rest between each group, interval to 50–60% HRmax exercise intensity slow walking, a total of 10 groups; relaxation for 5 min.	3 times/week	8 weeks	Stroop b, a
		MICT	12	5/7	12.73 ± 0.65	Warm-up 5 min; 30 min of continuous aerobic running at 65% HRmax; relaxation for 5 min.			
Mekari [32]	2020	HIIT	12	9/3	29 ± 10.3	Warm-up 5 min; power bike exercise at 100% PPO for 20 min, then rest for five minutes and perform two sets; 55 min total.	3 times/week	6 weeks	Stroop b
		MICT	13	9/4	35 ± 7.4	Warm-up 5 min; power cycling at 60% PPO for 35 min with a final rest of five minutes; total 45 min.			
Inoue [33]	2020	HIIT	10	-	30 ± 5.4	Warm-up 5 min; 10 interval runs on the treadmill (10 × 1:1-1 min at 100% MAV, interspersed with 1 min of passive recovery).	3 times/week	6 weeks	Stroop b
		MICT	10	-	30 ± 5.4	Warm-up 5 min; continuous running on the treadmill at 65% MAV for about 35 min.			
Wang [34]	2021	HIIT	28	-	11–12	Warm-up 3 min; basic stage 10 m fast folding run, open and closed jump, upgrade stage 10 m fast folding run, Bobbi jump, exercise intensity 85–90% HRmax; recovery 2 min.	3 times/week	8 weeks	Flanker b, a
		MICT	29	-	11–12	Warm-up 3 min; basic stage before and after the swing arm alternating jump, unarmed deep squat, upgrade stage prone knee lift, prone open jump, exercise intensity 60–69% HRmax; recovery 2 min.			
Li [35]	2021	HIIT	24	10/14	23.79 ± 1.82	Warm-up 5 min; 20 min of exercise in 1:1 interval mode (85% HRmax or higher intensity running exercise for 1 min with 1 min interval).	1 time		Stroop b
		MICT	24	12/12	23.13 ± 1.96	Warm-up 5 min; run for 20 min at 60–70% HRmax exercise intensity.			
Oliveira [36]	2021	HIIT	20	13/7	29.7 ± 8.3	Warm-up 5 min; then RPE was kept at 15–17 with 10 intervals on the treadmill (10 × 1, 1 min intervals with slow walking); recovery 5 min.	1 time		Stroop b, a
		MICT	21	10/11	33.2 ± 6.6	Warm-up 5 min; then RPE held at 13, 30 min on treadmill; recovery 5 min.			

Note: HRmax: Maximum Heart Rate; HRR: Heart Rate Reserve; VO_2_max: Maximum Oxygen Uptake; PPO: Peak Power Output; MAV: Maximum Aerobic Velocity; a: Correct Rate; b: Response Time; F: Female; M: Male.

## Data Availability

The original contributions presented in the study are included in the article, further inquiries can be directed to the corresponding author/s.

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
