# Peer review of "A Comparative Study of Inhibition Function between High-Intensity Interval Training and Moderate-Intensity Continuous Training in Healthy People: A Systematic Review with Meta-Analysis"

_ijerph, 2023, doi:10.3390/ijerph20042859_

Round 1
Reviewer 1 Report (New Reviewer)
Dear authors,
Thank you very much for the effort you put into your research. I would also like to thank you for giving me the opportunity to read and evaluate such a valuable research. I read your research in detail and really enjoyed reading it. Both the systematic and meta-analysis sections reveal the effort you put into your research. It is also a valuable research in terms of its subject and scope. That's why I think your research is ready for publication in its current form.
Yours sincerely
Author Response
Thank you very much for your valuable suggestions to this research before, and thank you very much for your recognition of this research.
Reviewer 2 Report (Previous Reviewer 1)
This manuscript was improved in the process of review.
Author Response
Thank you very much for your valuable advice, which makes our research more rigorous. Thank you again for your recognition of this study.
Reviewer 3 Report (Previous Reviewer 3)
Congratulations for the article again, The topic of the article is so interesting, I´m sure that the research applications will be of interest for this approach. I am very grateful for all your answers.
The only discordant point is the registration of the article, I leave it up to the editor to decide this last point.
Kinds regards.
Author Response
Thank you very much for your valuable advice, which makes our research more rigorous. We are very happy that you are interested in the content of this study. We will pay more attention to the registration of meta analysis in future studies, thank you again for your reminding.
Reviewer 4 Report (Previous Reviewer 4)
Thank you for your answers. I think the manuscript must be accepted.
Author Response
Thank you very much for your valuable advice, which makes the content of this study more in-depth and detailed. Thank you again for your recognition of this study.
Reviewer 5 Report (Previous Reviewer 6)
In this study, which was previously rejected and recommended to be submitted again, the authors compiled the effects of aerobic exercises performed at different intensities on cognitive functions and performed a meta-analysis. The authors made the correction that the exercise numbers should be written in the summary section, the correction that the inhibition function should be added to the introduction section of the literature summary, and many other corrections were made in the other sections (method and discussion). I am satisfied with these fixes. There is no harm in publishing it as it is. Congratulations to the authors.
Author Response
Thank you very much for your valuable comments on this study, which greatly improved the quality of our manuscripts. Thank you again for your recognition of this study.
This manuscript is a resubmission of an earlier submission. The following is a list of the peer review reports and author responses from that submission.
Round 1
Reviewer 1 Report
General Comments
This is an interesting meta-analysis, comparing the effects high-intensity interval training (HIIT) and moderate-intensity continuous training (MICT) on inhibition in executive function in healthy people. The Authors searched the PubMed, Science Direct, Web Of science, Cochrane, and CNKI databases for relevant articles on the inhibition function effects of HIIT and MICT in healthy populations. A total of 285 participants (adolescents, young and middle-aged, and elderly subjects) from 8 studies was included in this meta-analysis. Statistical analysis of the correct rate and response time indicators of the inhibitory function in the HIIT and MICT groups was performed. This meta-analysis clearly suggested that both HIIT and MICT could improve inhibition function in healthy people, whereas no significant difference in the improvement was noted.
The manuscript is generally well written. However, the design of this paper should be slightly improved before publishing.
1. In my opinion, it is obligatory to present in Abstract and 3.3. Research characteristics information about the age range and sex of the participants.
I suggest to replace the text from the beginning of the Discussion (Lines 70-89) to 1. Introduction because this is a literature overview rather than the analysis of the main results of this study.
The authors should present more limiting factors of this study at the end of the 4. Discussion. (1) The subjects with large age range (from adolescents to the elderly) were measured in this study. However, no age-related (or gender-related) differences were analysed. (2) No information regarding the habitual physical activity of the subjects was presented (trained, untrained, athletes, etc.), because this information is important for interpretation of the results of the studies.
Specific Comments
Abstract
Page 1. Please add the information about age range and sex (% male) of the participants.
3. Results
Page 5, 3.3. Research characteristics. Please add information about the age range and sex of the participants.
4. Discussion
Page 11, Lines 70-89. Please replace this text from the Discussion to 1. Introduction (see general Comments).
Page 12. Please add more limiting factors at the end of the Discussion (see General Comments).
Author Response
Our point-by-point responses for the questions of Reviewer1#
We are very thankful for the given comments and correction; we have followed exactly and carried out revision accordingly. Please see the revised manuscript, and the revisions are shown in red color as highlights.
- Page 1. Please add the information about age range and sex (% male) of the participants.
Response:
Thank you very much for your comments. Since some articles do not specify the ratio of male to female, we do not specify the specific number of male and female subjects in the abstract. But we added the ratio of men to women in the manuscript in Table 2 of the results section.
- Page 5, 3.3. Research characteristics. Please add information about the age range and sex of the participants.
Response:
Thank you very much for your comments. We added the male to female ratio of participants in Table 2 of the manuscript. In addition, the age information of the participants can also be found in Table 2.
3.Page 11, Lines 70-89. Please replace this text from the Discussion to 1. Introduction (see general Comments).
Response:
Thank you very much for your comments. We have revised the content of the discussion part in the manuscript into the introduction(Page 2, Lines 61-69), and added the analysis and discussion on the intervention population in the discussion(Page 14, Lines 121-139).
4.Page 12. Please add more limiting factors at the end of the Discussion (see General Comments).
Response:
Thank you very much for your comments. The information you mentioned about age, gender, and subjects' daily physical activities may affect the final result to some extent. We have taken your advice and added to the limitation section at the end of the manuscript how these factors might affect the outcome indicators(Page 15, Lines 164-168).

Reviewer 2 Report
Meta-analysis is very unique and hard work to do and so I appreciate your effort. However, 8 studies in total is not appropriate for meta-analysis as it cannot provide scientific relevance. Particularly, some meta-analysis were applied only by using two studies, which is not scientific and acceptable at all. Therefore, it is even beyond the major revision. Authors can try to change this study as a systematic review and publish somewhere else.
Author Response
Our point-by-point responses for the questions of Reviewer2#
We are very thankful for the given comments and correction; we have followed exactly and carried out revision accordingly. Please see the revised manuscript, and the revisions are shown in red color as highlights.
- Meta-analysis is very unique and hard work to do and so I appreciate your effort. However, 8 studies in total is not appropriate for meta-analysis as it cannot provide scientific relevance. Particularly, some meta-analysis were applied only by using two studies, which is not scientific and acceptable at all. Therefore, it is even beyond the major revision. Authors can try to change this study as a systematic review and publish somewhere else.
Response:
Thank you very much for your comments on the manuscript. As you said, we made a lot of efforts in the research process. We searched the databases of PubMed, Science Direct, Web Of science, Cochrane and CNKI, and were very careful and strict in the selection of articles. Finally, 8 articles were included in the meta-analysis. We cannot completely agree with you that 8 articles are not suitable for meta-analysis. Meta-analysis pays more attention to the quality of the articles and the content of the research. Some meta-analysis studies have included fewer documents, but they are very meaningful. We believe that our research should not be rejected just because the number of included articles is slightly less. In addition, go-nogo, stroop and flanker paradigms are commonly used to measure inhibition function, and the correct rate and response time are currently the most commonly used outcome indicators that can best reflect the quality of inhibition function. Therefore, we believe that it is appropriate to use these two as the final outcome indicators. Thank you again for your comments and we will consider including more outcome measures in similar studies in the future.

Reviewer 3 Report
ABSTRACT
- P. 1 line 21. You can also add some more statistics values.
- P. 1 line 23. You can also focus on the clinical application.
KEYWORDS
- P. 1 line 25. You can also include “healthy people” as a keyword.
INTRODUCTION
- It could be interesting to include more information about the prevalence or socio-demographic data of the inhibition function disruption in the different subjects.
- P. 2, line 47. Please, could you add a reference to the specified definition about HIIT?
- P. 2, line 53. The referencing style of the citation in the body of the manuscript must be reviewed (16,17,18).
- P 2, line 63. Please, could you add some references to the follow affirmation?
“Although some research has been conducted comparing the effects of HIIT and MICT methods on the improvement of inhibition function in executive functions in healthy individuals,”
- P 2, line 63. Do you know if previous authors have demanded more investigation with a higher strictness about this topic? If so, could you specify it?
- P 2, line 64. In my opinion, the objective of the study must be expressed in other way, the study is a systematic review with meta-analysis, not only a meta-analysis.
- P 2, line 66. Could you clarify how/why can you affirm that this study would provide a theoretical basis for future exercise practice and health interventions?
METHODS
In my opinion, is an important part for verifying the novelty of the review to register it previously in PROSPERO….
- P 3, line 107. Please, could you reference the used formula?
- P 3, line 112. The Cochrane tool is used to evaluate the risk of bias of the included studies but, in my opinion, other tool as PEDro or Down and Black should be included in order to evaluate the quality of the study.
RESULTS
- P 4, Figure 1. It could be interesting to use the new PRISMA flowchart.
- P 4, line 150. According to the Cochrane risk of bias assessment tool (19), I think that more than 1 study has high risk of bias.
- P 6, table 1. In my opinion, the table of the systematic review should include other information as % of male and female, the setting of the intervention, or the evaluated outcomes.
- P 9, figure 6,7,8. If the heterogeneity is lower than 50%, why a random model is applicated?
- P10, line 52. If the Egger method is applicated, it should be mentioned in the methods section.
DISCUSSION
- You could add a section to talk about the strength and the clinical application of your study.
Author Response
Our point-by-point responses for the questions of Reviewer3#
We are very thankful for the given comments and correction; we have followed exactly and carried out revision accordingly. Please see the revised manuscript, and the revisions are shown in red color as highlights.
1.P. 1 line 21. You can also add some more statistics values.
Response:
Thank you very much for your comments. We have revised the abstract to add the features of the included studies (Page 1, Line 18-20).
2.P. 1 line 23. You can also focus on the clinical application.
Response:
Thank you very much for your comments. We have supplemented this in the abstract of the manuscript (Page 1, Line 25-26).
3.P. 1 line 25. You can also include “healthy people” as a keyword.
Response:
Thank you very much for your advice, we have added “Healthy people” to the keywords(Page 1, Line 29).
4.It could be interesting to include more information about the prevalence or socio-demographic data of the inhibition function disruption in the different subjects.
Response:
Thank you very much for your comments. Your suggestion is of great research significance. However, since the subjects included in the study are healthy people, the disease status, exercise habits and other demographic indicators of the subjects were not involved in the included study. In the future, we may be able to conduct a new study from the perspective of how the disease disrupts people's inhibition function.
5.P. 2, line 47. Please, could you add a reference to the specified definition about HIIT?
Response:
Thank you very much for your comments. Thanks for your reflection, we have added references to the interpretation of the definition of HIIT in the manuscript [17].
[17] Li, Y. Effect of High-Intensity Interval Training on Different Training Populations. China Sport Science. 2015, 35, 59-75+96. Doi:10.16469/j.css.201508009
6.P. 2, line 53. The referencing style of the citation in the body of the manuscript must be reviewed (16,17,18).
Response:
Thank you very much for your comments. We have modified the way of article reference of 16, 17 and 18, because of the changes made to the manuscript, they now correspond to23, 24 and 25.
7.P 2, line 63. Please, could you add some references to the follow affirmation?
“Although some research has been conducted comparing the effects of HIIT and MICT methods on the improvement of inhibition function in executive functions in healthy individuals,”
Response:
Thank you very much for your comments. We have added references to this part of the manuscript [23,24,25].
[23] Kao, S. C. D. E., Ritondale, J. P. The acute effects of high-intensity interval training and moderate-intensity continuous exercise on declarative memory and inhibitory control. Psychology of Sport and Exercise. 2018, 38, 90-99.
[24] Coetsee, C., Terblanche, E. The effect of three different exercise training modalities on cognitive and physical function in a healthy older population. Eur Rev Aging Phys Act. 2017, 14, 13. Doi:10.1186/s11556-017-0183-5
[25] Zhang, Q., J. Effect of high-intensity interval training on brain executive function of young aged 12-14. Jinan, Shandong Normal University. 2020.
8.P 2, line 63. Do you know if previous authors have demanded more investigation with a higher strictness about this topic? If so, could you specify it?
Response:
Thank you very much for your comments. In the introduction of the manuscript, we describe and summarize the effects of HIIT and MICT on inhibition function. Kao [1] found that after one-time HIIT and MICT exercises, subjects were more accurate in performing Flanker tasks after HIIT than after MICT. Costsee's [2] study showed that MICT group was significantly better than HIIT group in improving executive function in the elderly. Qingju Zhang's [3] study found that the response of adolescents after HIIT and MICT intervention was reduced compared with that before exercise, but there was no significant difference between the two groups. In addition, Zhang Guoxiao et al. [4] showed that the physical fitness and inhibition ability of adolescents around 10 years old after HIIT and MICT intervention were improved compared with that before exercise, and the effect of HIIT was more significant. Fang Guoliang et al [5]. showed that HIIT had a more significant effect on improving cognitive function in the elderly.
[1] Kao, S. C. D. E., Ritondale, J. P. The acute effects of high-intensity interval training and moderate-intensity continuous exercise on declarative memory and inhibitory control. Psychology of Sport and Exercise. 2018, 38, 90-99.
[2] Coetsee, C., Terblanche, E. The effect of three different exercise training modalities on cognitive and physical function in a healthy older population. Eur Rev Aging Phys Act. 2017, 14, 13. Doi:10.1186/s11556-017-0183-5
[3] Zhang, Q. J. Effect of high-intensity interval training on brain executive function of young aged 12-14. Jinan, Shandong Normal University. 2020.
[4] Zhang, G. Effect of high intensity interval training on physical fitness and executive control ability of children. Jinan, Shandong Normal University. 2019.
[5] Fang, G., Zhang, L., Han, T., Zou, X., Zhang, H., Li, X., Wang, H., Shen, Y. Effects of High Intensity Interval Training on Congnitive Function in Older Adults. China Sport Science and Technology. 2020,56, 32-37. DOI:10.16470/j.csst.2020130
- P 2, line 64. In my opinion, the objective of the study must be expressed in other way, the study is a systematic review with meta-analysis, not only a meta-analysis.
Response:
Thank you very much for your comments. This study is a systematic review with meta-analysis, which we have revised in the manuscript (Page 2, Line 86).
10.P 2, line 66. Could you clarify how/why can you affirm that this study would provide a theoretical basis for future exercise practice and health interventions?
Response:
Thank you very much for your comments. Inhibition function refers to the ability of regulating our attention, emotion, behavior etc. by inhibiting internal and external interference, so as to effectively accomplish the expected task and goal [1]. Many studies have shown that exercise has a positive impact on cognitive function [2-3]. HIIT refers to the training method of carrying out several exercises lasting from a few seconds to a few minutes at a load intensity greater than or equal to the anaerobic threshold or the maximum lactate homeostasis, and the resting or low-intensity exercises are not sufficient for complete recovery between each two exercises [4]. It can induce greater peripheral vascular and cellular stress responses [5], and some researchers believe that HIIT has better physiological and health-related benefits than sustained aerobic exercise. However, some studies have shown that MICT can improve the inhibition function more significantly. In this study, meta-analysis was used to explore whether there were significant differences in the effects of HIIT and MICT on inhibitory function in a quantitative study. The results of this study show that the difference between the two exercise modes is not significant, and people should not be too attached to HIIT when choosing exercise intervention methods in the future. Although HIIT has attracted people's attention in recent years, its safety needs to be further observed and studied. Some subjects may withdraw from the experiment because they cannot bear the high exercise intensity. May not be applicable to all populations [6]. In the future, people should choose a more suitable way of exercise.
- Ahumada-Méndez, F., Lucero, B., Avenanti, A., Saracini, C., Muñoz-Quezada, M. T., Cortés-Rivera, C., Canales-Johnson, A. Affective modulation of cognitive control: A systematic review of EEG studies. Physiol Behav. 2022, 249, 113743. Doi:10.1016/j.physbeh.2022.113743
- Basso, J. C., Suzuki, W. A. The Effects of Acute Exercise on Mood, Cognition, Neurophysiology, and Neurochemical Pathways: A Review. Brain Plast. 2017, 2, 127–152. Doi:10.3233/BPL-160040
- Tian, S., Mou, H., Qiu, F. Sustained Effects of High-Intensity Interval Exercise and Moderate-Intensity Continuous Exercise on Inhibitory Control. Int J Environ Res Public Health. 2021, 18, 2687. Doi:10.3390/ijerph18052687
[4] Li, Y. Effect of High-Intensity Interval Training on Different Training Populations. China Sport Science. 2015, 35, 59-75+96. Doi:10.16469/j.css.201508009
[5] Gripp, F., Nava, R. C., Cassilhas, R. C., Esteves, E. A., Magalhães, C. O. D., Dias-Peixoto, M. F., de Castro Magalhães, F., Amorim, F. T. HIIT is superior than MICT on cardiometabolic health during training and detraining. Eur J Appl Physiol. 2021, 121, 159–172. Doi:10.1007/s00421-020-04502-6
[6] Oliveira, B. R., Slama, F. A., Deslandes, A. C., Furtado, E. S., & Santos, T. M. Continuous and high-intensity interval training: which promotes higher pleasure?. PloS One. 2013, 8, e79965. Doi:10.1371/journal.pone.0079965
11.In my opinion, is an important part for verifying the novelty of the review to register it previously in PROSPERO….
Response:
Thank you very much for your comments. This was an oversight on our part in not registering with PROSPERO in a timely manner. Thank you very much for your reminding. In future similar studies, we will definitely register relevant studies in advance.
12.P 3, line 107. Please, could you reference the used formula?
Response:
Thank you very much for your comments. We added a reference to the formula to the manuscript [26].
[26] Cumpston, M. S., McKenzie, J. E., Welch, V. A., Brennan, S. E. Strengthening systematic reviews in public health: guidance in the Cochrane Handbook for Systematic Reviews of Interventions, 2nd edition. J Public Health (Oxf), 2022, 44, e588–e592. Doi:10.1093/pubmed/fdac036
13.P 3, line 112. The Cochrane tool is used to evaluate the risk of bias of the included studies but, in my opinion, other tool as PEDro or Down and Black should be included in order to evaluate the quality of the study.
Response:
Thank you very much for your comments. In the manuscript, we added results that evaluated the quality of the included studies using PEDro. The results showed that one study had a score of 6 and was of high quality [25]. Six studies scored 5 [31,32,33,34,35,36]and one study scored 4 [24], which is of moderate quality (Table 1).
14.P 4, Figure 1. It could be interesting to use the new PRISMA flowchart.
Response:
Thank you very much for your comments. We have modified the PRISMA flow chart in the manuscript (Figure 1).
15.P 4, line 150. According to the Cochrane risk of bias assessment tool (19), I think that more than 1 study has high risk of bias.
Response:
Thank you very much for your comments. Five or more items satisfying the Cochrane bias risk assessment tool were considered low risk, 3-4 items satisfying moderate risk, and less than 3 items satisfying high risk. Among the eight studies, only one study met the 5 conditions [25], so it was low risk bias. Five studies met 4 of them [31,32,33,35,36], two studies met 3 of them [24,34], and these seven studies were medium risk bias.
16.P 6, table 1. In my opinion, the table of the systematic review should include other information as % of male and female, the setting of the intervention, or the evaluated outcomes.
Response:
Thank you very much for your comments. We have added the proportion of male and female subjects in Table 2. In addition, the specific interventions and the outcome indicators used can also be found in Table 2.
17.P 9, figure 6,7,8. If the heterogeneity is lower than 50%, why a random model is applicated?
Response:
Thank you very much for your comments. We examined the analytical methods in the manuscript. In the subgroup analysis in Figure 8, heterogeneity is less than 50% for both adolescents and adults, and a fixed-effects model should be used, which we modify in the manuscript. However, in the subgroup analysis in Figure 6 and 7, the heterogeneity of a group of people was higher than 50%, so we chose to use the random-effects model.
18.P10, line 52. If the Egger method is applicated, it should be mentioned in the methods section.
Response:
Thank you very much for your comments. The use of funnel plot and Egger method to check whether publication bias exists has been added to the section of statistical methods of manuscript (Page 4, Lines 170-171).
19.You could add a section to talk about the strength and the clinical application of your study.
Response:
Thank you very much for your comments. Thank you very much for reminding us to add the advantages of this research to the manuscript. We have supplemented the strengths and applications of this research in the manuscript (Page 14, Lines 140-154).

Reviewer 4 Report
Thank you for the invitation to review this manuscript.
It is a well-written manuscript, with a good methodology and results that may help exercise professionals to better understand the processes underlying inhibition function.
However, in my opinion, there is a missing section in the text that discusses the practical applications of these results and shows an example of one type of training with each of the two modalities HIIT and MICT. Figure 2 should also be modified to make it more comprehensible to the reader and, if possible, try to improve the tables and other figures in the manuscript to make them easier to visualise.
I congratulate the authors and hope that my comments will serve to improve the manuscript, which is already a very interesting manuscript.
Author Response
Our point-by-point responses for the questions of Reviewer4#
We are very thankful for the given comments and correction; we have followed exactly and carried out revision accordingly. Please see the revised manuscript, and the revisions are shown in red color as highlights.
1.However, in my opinion, there is a missing section in the text that discusses the practical applications of these results and shows an example of one type of training with each of the two modalities HIIT and MICT. Figure 2 should also be modified to make it more comprehensible to the reader and, if possible, try to improve the tables and other figures in the manuscript to make them easier to visualise.
Response:
Thank you very much for your comments. In the discussion section of the manuscript, we have added the advantages of this research and its practical application (Page 14, Lines 140-154). We believe that although HIIT has many benefits, compared with traditional MICT, its improvement in human inhibition function is not significantly different. In addition, HIIT has a great challenge to the human heart and lung function, and is not suitable for all people. Therefore, people should choose their own sports program, can not blindly follow the trend of sports. Figure 2 is the assessment of risk bias for the eight included studies, which is the most commonly used representation method. In the manuscript, we added a table that uses PEDro to evaluate the quality of the included studies, which is more intuitive than Cochrane. In addition, we also modified the table which summarized the 8 studies to make the table more complete and clear (Table 2). Other pictures are classic results presented in meta-analysis, so we did not change them in the manuscript.

Reviewer 5 Report
This is an interesting study about the effect of HIIT and MICT on inhibition function on healthy people. Th manuscript is overall well written and the hypothesis tested well documented. However the authors mention in the limitation section a few limitations. One of them, which is the major revision question I would like to address is that some studies might have been omitted in the literature review. The authors before attempting to publish a meta-analysis should be sure that have concluded all relevant studies in their analysis. I strongly suggest searching the literature more extensively and come up including all relevant studies to the best of their knowledge.
Author Response
Our responses for the questions of Reviewer5#
We are very thankful for the given comments and correction; we have followed exactly and carried out revision accordingly. Please see the revised manuscript, and the revisions are shown in red color as highlights.
1.This is an interesting study about the effect of HIIT and MICT on inhibition function on healthy people. Th manuscript is overall well written and the hypothesis tested well documented. However the authors mention in the limitation section a few limitations. One of them, which is the major revision question I would like to address is that some studies might have been omitted in the literature review. The authors before attempting to publish a meta-analysis should be sure that have concluded all relevant studies in their analysis. I strongly suggest searching the literature more extensively and come up including all relevant studies to the best of their knowledge.
Response:
Thank you very much for receiving your review comments. We agree with you that this is a very interesting study. We conducted multiple comprehensive searches in PubMed, Science Direct, Web Of Science, Cochrane and CNKI databases. We were very careful in the retrieval process and tried our best to avoid missing relevant articles. During the search, we found that not many articles directly compared inhibition control after high-intensity intermittent exercise and moderate-intensity continuous exercise. In order to be more rigorous, we excluded several of them because they were non-RCT studies, and only 8 of them met the inclusion criteria. Thank you again for your review, we have tried our best to find relevant studies.

Reviewer 6 Report
I make comments on the pdf version of your paper, however, in this style, this article cannot be published because intro and discussion sections does not sounds scientific, They needs to be improved and worked by all authors collectively. I suggest authors to improve its introduction and discussion section and resubmit.

Author Response
Our point-by-point responses for the questions of Reviewer6#
We are very thankful for the given comments and correction; we have followed exactly and carried out revision accordingly. Please see the revised manuscript, and the revisions are shown in red color as highlights.
- I make comments on the pdf version of your paper, however, in this style, this article cannot be published because intro and discussion sections does not sounds scientific, They needs to be improved and worked by all authors collectively. I suggest authors to improve its introduction and discussion section and resubmit.
Response:
Thank you very much for taking time out of your busy schedule to review our manuscript. Based on your suggestions and those of other reviewers, we have revised the introduction, result and discussion of the manuscript. In the introduction, we put the previous discussion about the effect of exercise on inhibition function into the introduction(Page 2, Lines 51-69). In the results section, we added a new table (Table 1), using PEDro to evaluate the literature quality of the included studies. In the discussion section, we supplement the discussion of subgroup analysis of different populations and the advantages of this study (Page 14, Lines 121-154). We hope you can give us your valuable comments on the revised manuscript.
